# Transformer-Based Detection for Highly Mobile Coded OFDM Systems

**DOI:** 10.3390/e25060852

**Published:** 2023-05-26

**Authors:** Leijun Wang, Wenbo Zhou, Zian Tong, Xianxian Zeng, Jin Zhan, Jiawen Li, Rongjun Chen

**Affiliations:** 1School of Computer Science, Guangdong Polytechnic Normal University, Guangzhou 510665, China; wangleijun@gpnu.edu.cn (L.W.); a1402057929@gmail.com (W.Z.); t15119803678@gmail.com (Z.T.); gszhanjin@gpnu.edu.cn (J.Z.); lijiawen@gpnu.edu.cn (J.L.); 2Guangdong Provincial Key Laboratory of Big Data Computing, The Chinese University of Hong Kong, Shenzhen 518000, China

**Keywords:** transformer, deep neural network (DNN), deep learning, orthogonal frequency division multiplexing (OFDM), high-speed railway, wireless communication

## Abstract

This paper is concerned with mobile coded orthogonal frequency division multiplexing (OFDM) systems. In the high-speed railway wireless communication system, an equalizer or detector should be used to mitigate the intercarrier interference (ICI) and deliver the soft message to the decoder with the soft demapper. In this paper, a Transformer-based detector/demapper is proposed to improve the error performance of the mobile coded OFDM system. The soft modulated symbol probabilities are computed by the Transformer network, and are then used to calculate the mutual information to allocate the code rate. Then, the network computes the codeword soft bit probabilities, which are delivered to the classical belief propagation (BP) decoder. For comparison, a deep neural network (DNN)-based system is also presented. Numerical results show that the Transformer-based coded OFDM system outperforms both the DNN-based and the conventional system.

## 1. Introduction

Orthogonal frequency division multiplexing (OFDM) shows great performance in the frequency-selective fading channels. However, in the highly mobile communication system, the Doppler effect destroys the orthogonality among the subcarriers of the OFDM system, resulting in intercarrier interference (ICI), which will degrade the system’s performance. In order to mitigate ICI in the mobile OFDM system, an equalizer or detector, e.g., a zero forcing (ZF) detector or minimum mean square error (MMSE) detector, should be implemented. To further combat channel distortion, it is necessary to use a channel coding scheme. Thus, a soft demapper is needed between the detector and the channel decoder. We have already studied the mobile coded OFDM system with a flexible coding scheme called block Markov superposition transmission (BMST) [1,2]. In this paper, we still focus on the mobile coded OFDM system.

In recent years, deep learning (DL) has achieved great success in various fields, such as computer vision [3,4,5,6], natural language processing, speech recognition, trajectory prediction [7]. Therefore, researchers in wireless communication are seeking to applying DL to various aspects of communication to achieve good performance. Currently, research in this area has already shown a trend of combining wireless transmission with deep learning in the physical layer, but all studies are still in the initial exploration stage [8,9,10].

In [11], a deep neural network (DNN) was utilized to solve the problem of channel estimation and signal detection in the OFDM system. In [12], the receiver uses the existing architecture to perform channel estimation and signal detection separately, and the authors combine each module with deep learning to improve the performance. In [13], the proposed receiver is trained with both offline and real-time online data to capture channel characteristics that are ignored during offline training. In addition, the deep learning approach has been introduced to signal detection in OFDM with index modulation (OFDM-IM) [14]. In particular, a Transformer-based detector was utilized to the OFDM-IM in [15], and a heterogeneous Transformer-based device activity detection method was proposed for the modern machine-type communications [16]. The Transformer [17], which is a network structure based on a self-attention mechanism, was proposed in 2017. Shortly thereafter, it showed remarkable effects in natural language processing [18], and as it demonstrated remarkable effects in computer vision, its application gradually became an important aspect of deep learning.

In this paper, we propose a Transformer-based detection/demapping algorithm for the mobile coded OFDM system. Although there are some studies on deep learning for channel coding [19] and decoding [20,21,22,23], the performance is not satisfactory when decoding directly via DL, especially regarding the long capacity-approaching codes. In this paper, we employ the low-density parity check (LDPC) codes, and the soft information given to the conventional belief propagation (BP) decoder is computed by deep learning. The main contributions of this paper are as follows:A Transformer-based detection algorithm is proposed for the coded OFDM system. Although DL-based detectors do not outperform the conventional detector in the uncoded OFDM system, they have better performance in the coded OFDM system.In our coded OFDM system, the LDPC codes are performed, and the soft information is required by the decoder. Thus, we propose the soft demapping algorithm based on Transformer.In the OFDM system, it is difficult to compute the mutual information based on the optimal detector. Thus, we can compute the mutual information with a suboptimal detector, which can be regarded as the soft information quality (SIQ) [24,25]. We compute the SIQ with the assistance of the Transformer network.

The remainder of the paper is organized as follows. Section 2 introduces the overall system model, including some classical detection algorithms. Section 3 proposes the Transformer-based detection and demapping algorithm. Section 4 gives the experimental results. Finally, Section 5 concludes the paper.

## 2. System Model

### 2.1. The Coded OFDM System Model

In this section, we introduce a single-antenna coded OFDM system that utilizes *N* subcarriers over doubly selective channels. Figure 1 depicts the block diagram of the general coded OFDM system. At the transmitter, the information bit stream  $\underline{u}$ is first encoded by the encoder to produce a coded sequence $\underline{c}$, which will be mapped to the vector sequence $\underline{x}$. The sequence c, which represents a block of the sequence $\underline{c}$, is then mapped to the vector of *N*-coded frequency-domain symbols, denoted as $x={\left[x_{0},\cdots ,x_{N-1}\right]}^{T}$. Each $x_{i}$ in the vector represents the symbol transmitted on the *i*th subcarrier of one OFDM symbol, chosen from a complex signal constellation S with $\left|S\right|=2^{M_{c}}$, in which $M_{c}$ represents the bit number carried by one symbol, and we assume that the average symbol energy $E[|x_{i}{|}^{2}]=1$.

The time-domain symbols are obtained by using the inverse discrete Fourier transform (IDFT), which can be implemented through the inverse fast Fourier transform (IFFT). A cyclic prefix (CP) or guard interval of length $N_{cp}\geq N_{tap}-1$ is added to the beginning of the time-domain symbols. The resulting time-domain symbols are then transmitted over the doubly selective channel, which has $N_{tap}$ taps.

After the receiver removes the CP and applies the discrete Fourier transform (DFT), the receive vector can be expressed as
(1)$$y={\mathrm{FH}}_{t}F^{H}x+Fw,$$
where y, x, and w are the sub-blocks of $\underline{y}$, $\underline{x}$, and $\underline{w}$, respectively, corresponding to a single OFDM symbol. F represents the unitary DFT matrix of size *N*. $H_{t}$ is the time-domain channel matrix, whose construction will be discussed in detail in the following section. The vector w denotes the time-domain additive white Gaussian noise (AWGN), whose entries are independently and identically distributed (i.i.d.) according to $\mathrm{CN}(0,\sigma^{2})$. Then, the signal-to-noise ratio (SNR) observed at the receiver can be defined as SNR = $1/\sigma^{2}$.

Denoting the frequency-domain matrix as $H_{f}={\mathrm{FH}}_{t}F^{H}$, the receive vector can be expressed as
(2)$$y=H_{f}x+w_{f}.$$

The frequency-domain noise $w_{f}$ has the same statistical properties as the time-domain noise w due to the unitary matrix property.

### 2.2. The Channel Model

As mentioned earlier, doubly selective fading channels are used in high-speed railway communication systems. This study assumes that the receiver has access to channel state information (CSI). In the OFDM system, the discrete-time channel matrix element is represented by the discrete-time impulse response $h_{n,m}$, where *n* is the subcarrier index, and *m* is the tap index, with 0≤n≤N−1 and $0\leq m\leq N_{tap}-1$. The Jakes’ Doppler spectrum [26] enables the calculation of $h_{n,m}$ as follows:(3)$$h_{n,m}=\sqrt{\frac{P_{m}}{4M}}\left\{\sum_{i=1}^{M}\sqrt{2}e^{j\psi_{i}}\left[e^{j(\omega_{i}n+\phi )}+e^{-j(\omega_{i}n+\phi )}\right]\right\}$$
with $\omega_{i}=\omega_{d}\cos\alpha_{i}$, where
$$\omega_{d}=\frac{2\pi \vartheta_{\max}}{N},$$
$$\alpha_{i}=\frac{2\pi i-\pi +\theta }{4M},\;\;\;i=1,2,\cdots ,M.$$In this paper, let *M* be the number of sinusoids (M=32). $P_{m}$ denotes the power-delay profile (PDP), where $\vartheta_{max}$ represents the normalized maximum Doppler shift. The statistical variables θ, ϕ, and ψ are independent and uniformly distributed over the interval [−π,π) for all *i*. The time-domain matrix $H_{t}$ contains non-zero elements generated using Equation (Equation 3). Given the OFDM system structure, $H_{t}$ can be expressed as follows:(4)$$H_{t}=L+U.$$Equation (Equation 4) involves a N×N lower triangular matrix whose nonzero element $L_{n,m}=h_{n,n-m},0\leq m\leq n\leq N-1$, and U is a N×N upper triangular matrix whose nonzero element $U_{n,m}=h_{n,N+n-m},0\leq n\leq N_{tap}-1,N-N_{tap}+1\leq m\leq N-1,n\leq m$.

### 2.3. The Classical Signal Detection and Demapping Algorithm

When we obtain the receive vector y with optimal maximum likelihood (ML) detection, the transmitted vector x can be obtained as follows:(5)$$\hat{x}=\arg\max_{x\in S^{N}}f\left(y|x,H_{f}\right),$$
where
(6)$$f\left(y|x,H_{f}\right)={\left(\frac{1}{\pi \sigma^{2}}\right)}^{N}\exp\left(-\frac{∥y-H_{f}{x∥}^{2}}{\sigma^{2}}\right).$$In Equation (Equation 5), the size of the symbol set S and the number of OFDM subcarriers *N* have a great impact on the complexity, especially when using a large number of subcarriers, as the optimal detection has little practical value. Therefore, it is necessary to perform an equalization operation on the received signal to eliminate the ICI before detection. When the receive vector y—see Equation (Equation 2)—is obtained, the process of equalization can be described as follows:(7)$$Gy=GH_{f}x+Gw_{f},$$
where G is the preprocessing matrix. When the ZF detector is used, Equation (Equation 7) can be rewritten as
(8)$$H_{f}^{†}y=x+H_{f}^{†}w_{f},$$
where $H_{f}^{†}={\left({H_{f}}^{H}H_{f}\right)}^{-1}{H_{f}}^{H}$ is the pseudoinverse of the frequency-domain matrix $H_{f}$.

When MMSE equalization is used, the preprocessing matrix
(9)$$G={\left(H_{f}^{H}H_{f}+\sigma^{2}I\right)}^{-1}H_{f}^{H}.$$

In the uncoded system, the vector $\tilde{y}=\mathrm{Gy}$ can be used to estimate the vector x by
(10)$${\hat{x}}_{i}=\arg\min_{x_{i}\in S}{∥{\tilde{y}}_{i}-x_{i}∥}^{2},$$
where ${\tilde{y}}_{i}$ and $x_{i}$ are the *i*th elements of the vectors $\tilde{y}$ and x, respectively. However, in the coded OFDM system, soft information is required by the demapper/decoder. For any symbol $x_{i}$ transmitted on one subcarrier, the soft information is computed as
(11)$$f\left({\tilde{y}}_{i}|x_{i}\right)\propto \exp\left(-\frac{|{\tilde{y}}_{i}-x_{i}{|}^{2}}{∥g_{i}{∥}^{2}\sigma^{2}}\right),$$
where $g_{i}$ represents the *i*th row of the preprocessing matrix $G={\left[g_{0},g_{1},\cdots ,g_{N-1}\right]}^{T}$. In the following section, we will describe the DL (Transformer)-based network, which can realize the functions of a soft demapper and detector; see Figure 1. It is important to note that the long LDPC codes are mainly used in our following experiments, so the codeword set is too large. From the observation of the simulation based on our proposed network, the DL-based decoders do not have good performance. Thus, we still use the conventional BP decoder in this paper.

### 2.4. Rate Allocation

In the coded modulation system, in order to determine the code rate of the coding scheme, it is necessary to compute or simulate the mutual information. However, as mentioned above, it is difficult to compute the likelihood probability in the OFDM system, resulting in the failure to solve the mutual information. Thus, we can compute SIQ, the mutual information, with the detector. Assume that each subcarrier sends an *M*-ary QAM symbol, and let *X* and $\tilde{Y}$ be the corresponding random vectors for the sending symbol *x* and receiving preprocessed symbol *y*, respectively. Obviously, *X* is a discrete random variable and $\tilde{Y}$ is a continuous random variable. Assuming that *X* follows a uniform distribution and the channel matrix H is known, the SIQ can be computed as
(12)$$I\left(X;\tilde{Y}|H\right)=\log_{2}\left(M\right)+E\left[\sum_{X}f\left(x_{i}|y_{i}^{\prime },H\right)\log_{2}f(x_{i}|y_{i}^{\prime },H)\right],$$
in which the probability $f(x_{i}|y_{i}^{\prime },H)$ can be computed by the classical demapping algorithm described above, or by the output of the DL network.

## 3. The Detection and Soft Demapping Algorithm with Deep Learning

### 3.1. Transformer

The traditional Transformer is essentially an encoder–decoder structure, which can be divided into two parts, the encoder and the decoder, as shown in Figure 2.

The left part is the encoding component, mainly composed of multi-head attention mechanisms and feed-forward neural networks. Encoding has two sub-layers: one is the multi-head attention mechanism layer, which uses self-attention mechanisms to learn the relationships between different dimensions of the data; the other is the forward propagation layer, which is a simple fully connected layer that performs the same operation on each position vector, including linear transformation and activation functions, and then produces encoding, which is passed to the encoding layer.

The right part is the decoding component, which is mainly composed of masked self-attention mechanisms and feed-forward neural networks. There are three sub-layers in the decoding layer, two of which are multi-head attention mechanism layers. The lower attention mechanism layer learns the relationships within the data using self-attention mechanisms, and then the layer outputs the results together with the results transmitted by encoding to the attention layer above. The attention layer calculates the correlation between the encoding and decoding, and can explore the relationship between the input sequence and the target sequence.

### 3.2. Experimental Method with Transformer

As shown in Figure 1, we treat the signal detection and demapping processes as a black box and replace them entirely with a Transformer network.

The detailed Transformer structure designed for the OFDM system is depicted in Figure 3. The Transformer network receives a time-domain frame $\left(\underline{y},\underline{h}\right)$ as input and recovers the transmitted bit data or probability in an end-to-end manner. The network is trained offline and deployed online. During the training phase, the transmitted bits are randomly generated as labels and modulated with channel information to form frames. A simulated channel is generated using a specific channel model, which varies with each frame. In the deployment phase, no equalization or detection is required, and the trained parameters are directly applied to achieve end-to-end bit recovery.

The designed Transformer model is suitable for encoding and non-encoding scenarios. For non-encoding signal detection, the input consists of the received signal $\underline{y}$ and the CSI $\underline{h}$, which serve as features. The transmitted bits or symbols are used as labels and, after passing through the Transformer, output the hard decision of the transmitted bits or the symbol probabilities using the *softmax* function. The symbol probabilities can be used to compute the SIQ.

In the case of encoding, the labels for the network are the encoded bits, and the network outputs the bit probabilities, which are computed by the *softmax* function. The probabilities are then sent to the BP decoder to recover the original information bits.

### 3.3. Model Training

The Transformer network structure is related to the input and output dimensions; see Figure 3. The Transformer model used consists of $N_{t}$ Transformer layers, each of which is composed of an embedding layer, a Transformer layer, and a fully connected layer. The Transformer consists of NDQLinear, Linear, LayerNorm, and Dropout layers, with neuron numbers of 512, 512, and 256, respectively. The number of input layer neurons corresponds to the sum of the real and imaginary parts of two OFDM signals, and the output corresponds to 16 bits. Since the data signal is modulated using QPSK with 64 effective subcarriers and a total of 128 bits, 8 independent networks are required, which are then concatenated for the final output. The hidden layers of the network use ReLU as the activation function.

The parameters in the Transformer network are trained using the training data, with the goal of minimizing the difference between the output of the neural network and the label data. The difference can be described in various ways, and this study uses cross-entropy loss. The cross-entropy between two probability distributions $P(x_{i})$ and $Q(x_{i})$ for a random variable *X* can be computed as
(13)$$H\left(P,Q\right)=-\sum_{i=1}^{n}P\left(x_{i}\right)\log Q(x_{i}),$$
which can measure the difference between the two distributions.

Taking encoded data as an example, y represents the received data at the receiving end, h represents the CSI data, and the output encoded data
(14)$$\underline{\hat{c}}=Transformer\left(\underline{y},\underline{h}\right).$$The goal is to find the best set of weights and biases of the network, denoted by $\theta^{*}$, that minimize the loss function, which is defined as follows:(15)$$\theta^{*}=\arg\min_{\theta }Loss\left(\underline{c},\underline{\hat{c}}\right),$$
in which $\underline{c}$ represents the label.

### 3.4. DNN Detection Algorithm

DNNs, also known as multi-layer perceptrons, consist of input layers, several hidden layers, and output layers; see Figure 4. Each hidden layer contains several neurons that do not interfere with each other, and they connect to adjacent layers. A single neuron multiplies each input by the corresponding weight and adds the bias parameter, finally reaching the output layer through a non-linear activation function. The core of a DNN is that it can perform self-optimization through back-propagation, but as the number of layers and neurons increases, the training will face problems such as gradient disappearance, slow convergence, and local minimum values. In order to improve the training speed and reduce the computational complexity, the classic gradient descent (GD) method has been replaced by stochastic gradient descent, which randomly selects data to calculate each loss and gradient. However, actual execution may be very slow because the entire data set needs to be traversed. Therefore, a commonly used approach is to randomly extract a subset of samples for training each time updates are calculated, which is called mini-batch stochastic gradient descent. However, these algorithms may still converge to local optimal solutions.

The DNN network can be used not only for channel estimation, equalization, and detection in traditional OFDM systems, but also for decoding. In this paper, the DNN network is built for comparison with the Transformer. The signal $\underline{y}$ and the CSI $\underline{h}$ are fed into a DNN network, with the same set of parameters for each network. The hidden layers of the DNN network are set to 256, 512, and 1024, respectively. The output of the two networks are concatenated together and passed through a fully connected layer and then a hidden layer of 256 before outputting 8 values, which are used to calculate the cross-entropy and probability for two dimensions. There are a total of 32 networks, with each network outputting 4 values. The 32 network outputs are merged together to obtain the complete output. The methods for calculating the probabilities and hard decision in the DNN network are identical to those in the Transformer network described above and will not be reiterated here.

## 4. Experimental Results

In this section, the numerical examples are based on Monte Carlo simulation over time-varying frequency-selective Rayleigh fading channels, and the PDP is $P_{i}=\alpha e^{-0.6i}$, $0\leq i\leq N_{tap}-1$, where α is a normalization constant. The relation between the normalized maximum Doppler shift $\vartheta_{\max}$ and the relative speed between the transmitter and receiver υ is $\vartheta_{\max}=\frac{f_{c}}{F_{c}}\frac{\upsilon }{c_{0}}$. The simulation parameters are shown in Table 1.

In the following simulations, we mainly use regular LDPC codes, which are constructed by the progressive-edge-growth (PEG) algorithm. The sum product algorithm (SPA) is employed for the decoding of the LDPC codes, in which the maximum iteration number is 40.

**Example 1.** 
*In this example, the bit-error-rate (BER) performance of the uncoded OFDM system with the variety detector is as depicted in Figure 5. The training data are generated when SNR = 20 dB in the DL simulation. From the figure, we have the following observations.*

*The BER performance of the uncoded OFDM system with the conventional detectors is better than that with the DL detectors in the high SNR region.*

*The MMSE detector performs better than the ZF detector, while the Transformer-based detector performs better than the DNN-based detector.*



**Figure 5 entropy-25-00852-f005:**
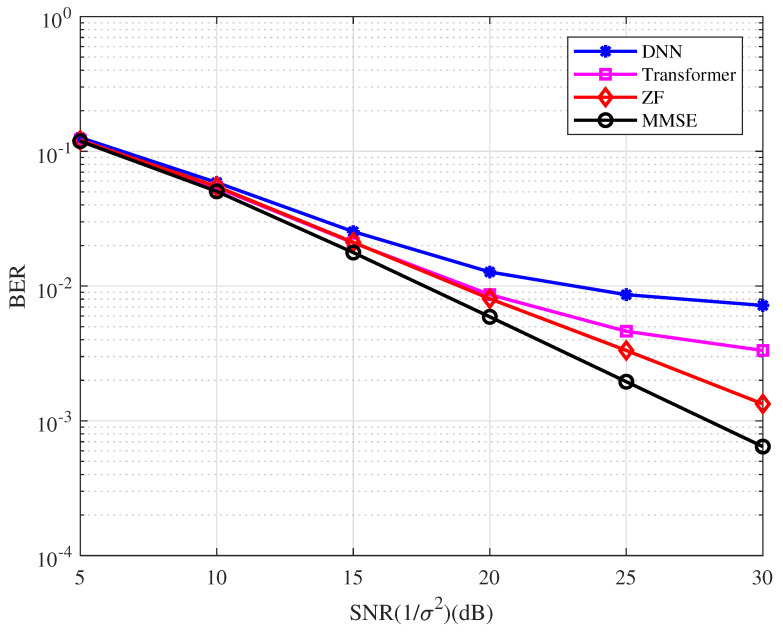
BER performance of the coded OFDM system with the different types of detector.

**Example 2.** *In this example, the mutual information based on the type of detector, which can be regarded as SIQ, versus the SNR is as depicted in Figure 6. For the ZF detector, we can observe that the Shannon limit for QPSK to achieve the spectral efficiency of 0.75, 1.0, 1.25, 1.5 bits/symbol/subcarrier is approximately −0.1, 2.3, 4.7, 7.5 dB, respectively. With the probability generated by the* softmax *function for the DNN and Transformer, their SIQs are higher than that of the conventional detector in the high SNR region. In the following examples, we take the ZF Shannon limit as a benchmark for comparison.*

**Example 3.** 
*In this example, a ${\left[8,3\right]}^{128}$ regular LDPC code is used. The BER performance of the coded OFDM with the type of detector at 0.75 bits/symbol/subcarrier spectral efficiency is depicted in Figure 7. From the figure, we have the following observations.*

*At the BER of $10^{-4}$, the BER performance with the ZF detector is approximately 6.2 dB away from the Shannon limit.*

*The BER performance corresponding to the Transformer detector has an approximately 2.0 dB gain compared with the DNN detector.*

*The Transformer-based system performs better than the ZF system in the high SNR region.*

*The ZF detector has better performance than the MMSE detector in the coded OFDM system.*



**Figure 7 entropy-25-00852-f007:**
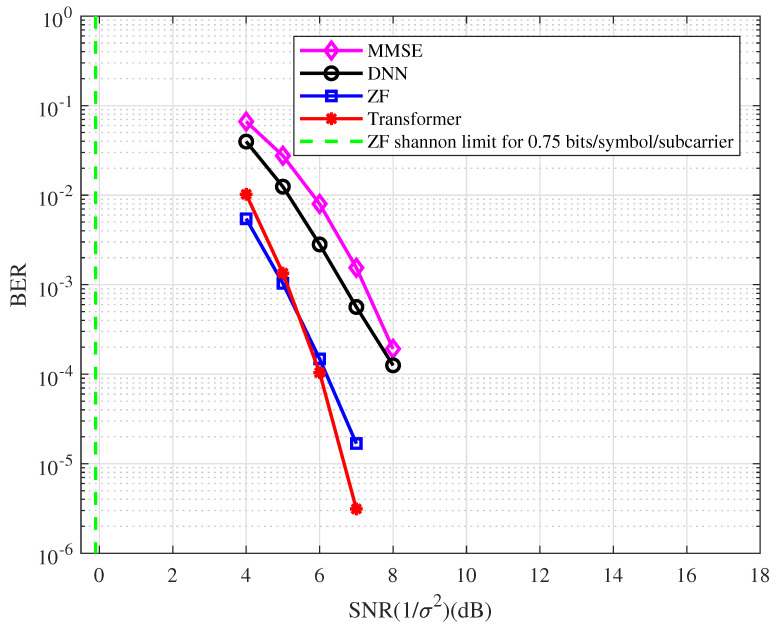
The BER performance of the coded OFDM with the different types of detector at 0.75 bits/symbol/subcarrier spectral efficiency.

**Example 4.** 
*In this example, a ${\left[2,1\right]}^{512}$ regular LDPC code is used. The BER performance of the coded OFDM with the type of detector at 1.0 bits/symbol/subcarrier spectral efficiency is depicted in Figure 8. From the figure, we have the following observations.*

*At the BER of 10−4, the BER performance with the ZF detector is approximately 6.0 dB away from the Shannon limit.*

*At the BER of $10^{-4}$, the BER performance corresponding to the Transformer detector has an approximately 0.4 dB, 1.4 dB, and 2.0 dB gain compared with the ZF, DNN, and MMSE detectors, respectively.*

*The BER performance with the ZF detector has an approximately 1.6 dB gain compared with the MMSE detector in the coded OFDM system.*



**Figure 8 entropy-25-00852-f008:**
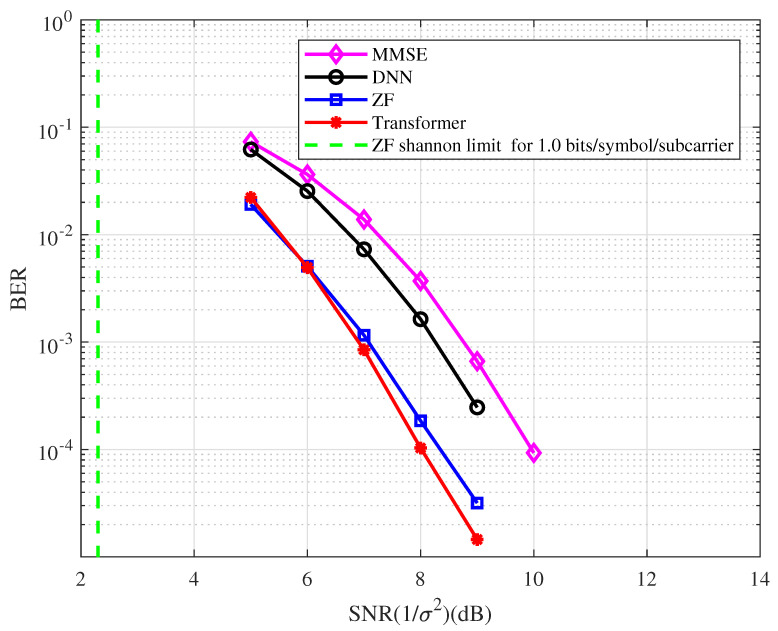
The BER performance of the coded OFDM with the type of detector at 1.0 bits/symbol/subcarrier spectral efficiency.

**Example 5.** 
*In this example, a ${\left[8,5\right]}^{128}$ regular LDPC code is used. The BER performance of the coded OFDM with the type of detector at 1.25 bits/symbol/subcarrier spectral efficiency is depicted in Figure 9. From the figure, we have the following observations.*

*At the BER of $10^{-4}$, the BER performance with the ZF detector is approximately 6.0 dB away from the Shannon limit.*

*The Transformer-based system performs better than the ZF system in the high SNR region.*

*At the BER of $10^{-4}$, the BER performance corresponding to the Transformer detector has an approximately 1.0 dB and 1.4 dB gain compared with the DNN and MMSE detectors, respectively.*

*The BER performance with the ZF detector has an approximately 1.1 dB gain compared with the MMSE detector in the coded OFDM system.*



**Figure 9 entropy-25-00852-f009:**
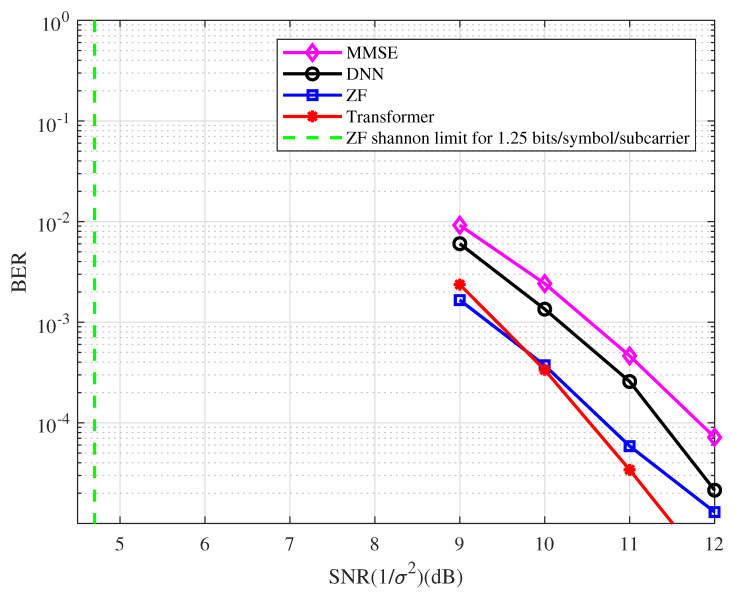
The BER performance of the coded OFDM with the type of detector at 1.25 bits/symbol/ subcarrier spectral efficiency.

**Example 6.** 
*In this example, a ${\left[4,3\right]}^{256}$ regular LDPC code is used. The BER performance of the coded OFDM with the type of detector at 1.5 bits/symbol/subcarrier spectral efficiency is depicted in Figure 10. From the figure, we have the following observations.*

*At the BER of $10^{-4}$, the BER performance with the ZF detector is approximately 7.0 dB away from the Shannon limit.*

*The DL-based system performs better than the ZF system in the high SNR region.*

*At the BER of $10^{-4}$, the BER performance corresponding to the Transformer detector has an approximately 0.8 dB gain compared with the MMSE detector.*



**Figure 10 entropy-25-00852-f010:**
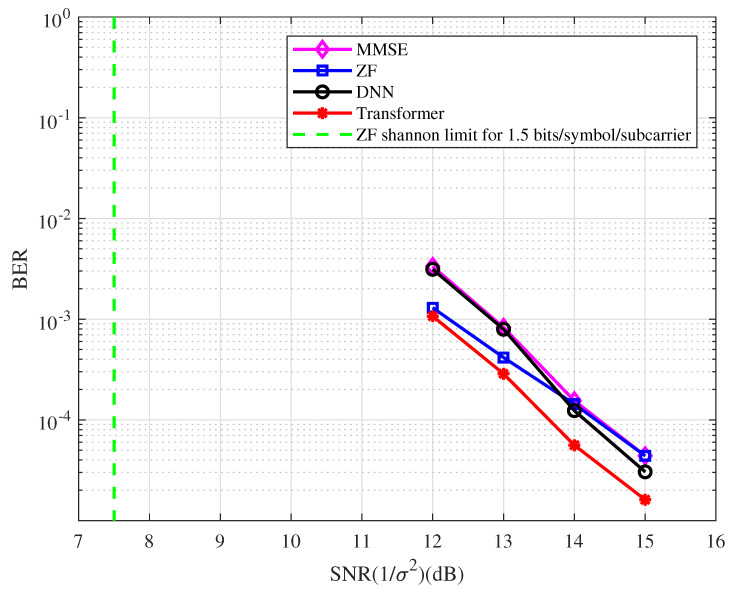
The BER performance of the coded OFDM with the type of detector at 1.5 bits/symbol/ subcarrier spectral efficiency.

From the examples described above, we can observe that the conventional detector performs better than the DL-based detector in the uncoded system, while the situation is the opposite in the coded system with the BP decoding algorithm based on the Tanner graph.

An interesting question is whether the system employing our network still has good performance when decoding without BP based on the Tanner graph. We present an example for comparison.

**Example 7.** 
*In this example, a (3,2,2) convolutional code (CC) is used. The CC is defined by the generation matrix $\left(\begin{array}{ccc} 1+D & D & 1+D\\ D & 1 & 1 \end{array}\right)$. We use the Bahl–Cocke–Jelinek–Raviv (BCJR) decoding algorithm. The BER performance of the CC-coded OFDM with the type of detector is depicted in Figure 11. From the figure, we have the following observations.*

*As with the uncoded system, the MMSE detector performs better than the ZF detector.*

*The DL-based systems perform no better than the ZF and MMSE systems and have an error floor in the high SNR region. This implies that the proposed network is more suitable for the decoding algorithm based on the factor graph.*



**Figure 11 entropy-25-00852-f011:**
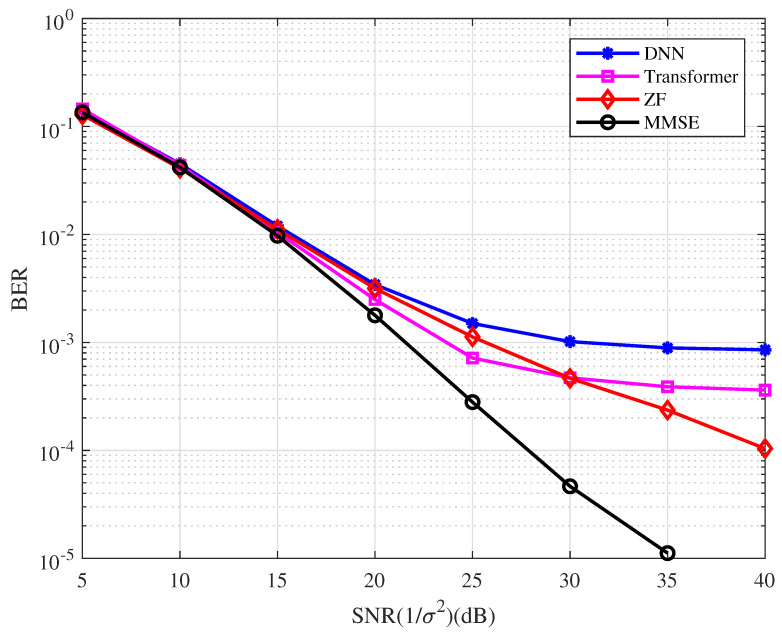
The BER performance of the CC-coded OFDM with the type of detector.

## 5. Conclusions

In the high-speed railway wireless communication system, an equalizer or detector, e.g., the ZF or MMSE detector, should be used to mitigate the intercarrier interference (ICI). In our schemes, the LDPC codes are employed to improve the error performance. Thus, a soft demapper should be designed for message delivery. In this paper, a Transformer-based detector/demapper is proposed in the mobile coded OFDM system. The proposed network can compute the symbol or bit probabilities, which can be used to calculate the SIQ or deliver it to the decoder. We also designed a DNN-based detector/demapper for comparison. The BER performance of the uncoded system and coded systems with different LDPC codes was presented. Numerical results show that although the Transformer-based uncoded OFDM systems do not outperform the systems utilizing the ZF or MMSE detectors, they have better performance in the coded OFDM system compared with both the conventional and DNN-based systems.

## Figures and Tables

**Figure 1 entropy-25-00852-f001:**
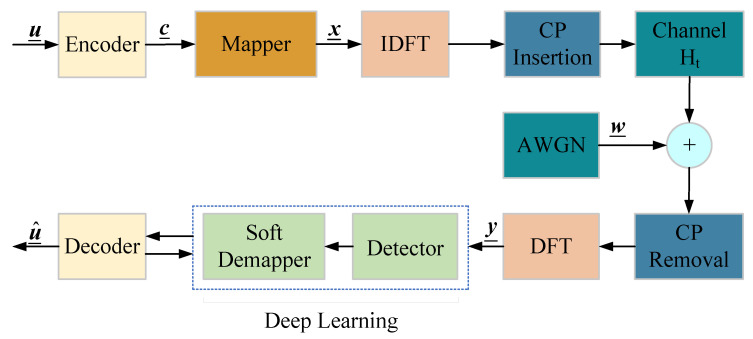
The block diagram of the coded OFDM system.

**Figure 2 entropy-25-00852-f002:**
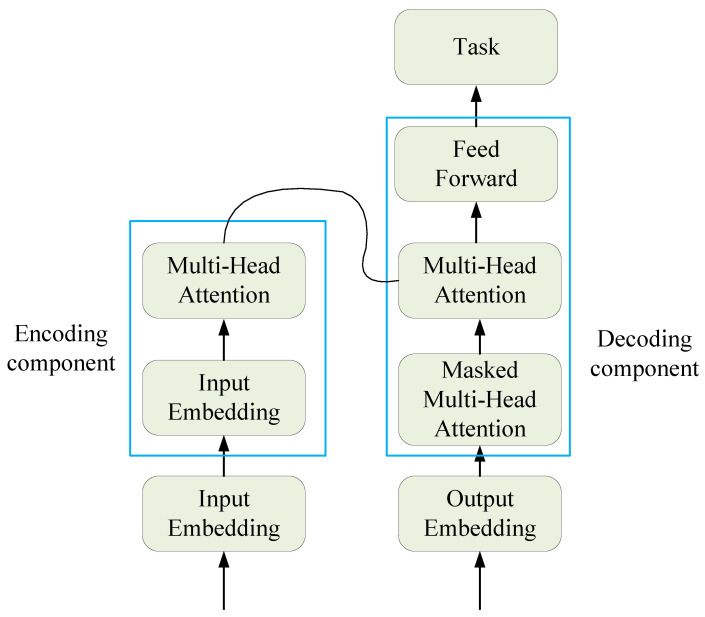
The block diagram of the traditional Transformer structure.

**Figure 3 entropy-25-00852-f003:**
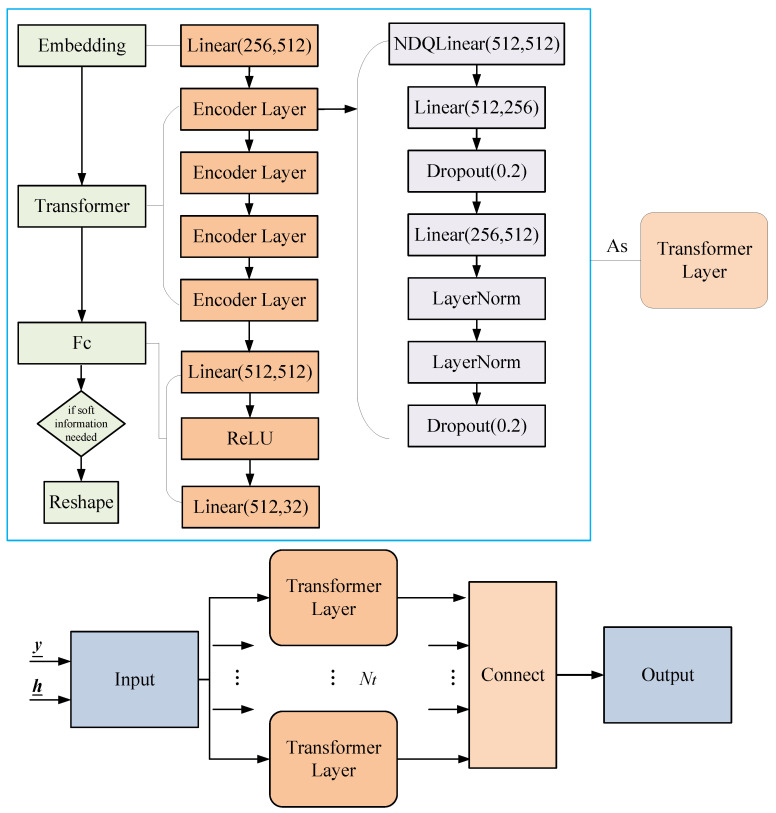
The block diagram of the designed Transformer structure for the OFDM system.

**Figure 4 entropy-25-00852-f004:**
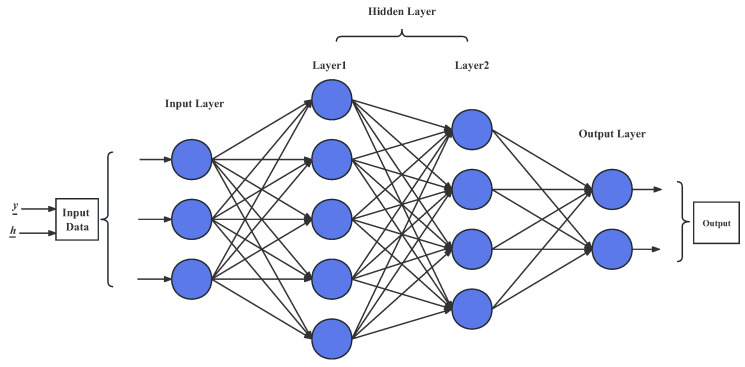
The DNN model for the OFDM system.

**Figure 6 entropy-25-00852-f006:**
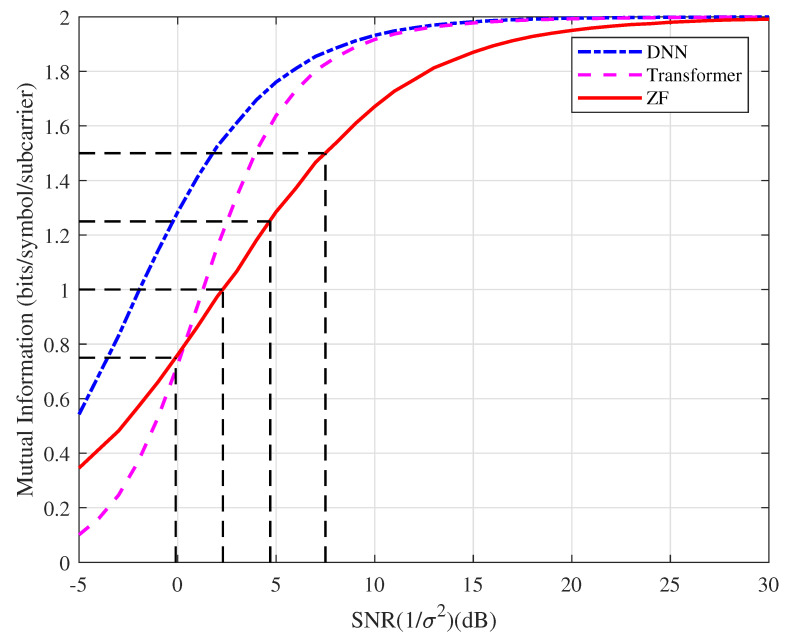
Mutual information based on the type of detector (SIQ) by Monte Carlo simulation.

**Table 1 entropy-25-00852-t001:** Simulation parameters.

Parameter	Value
Subcarriers *N*	64
Subcarrier Spacing $F_{c}$	15 KHz
Carrier Frequency $f_{c}$	2 GHz
Multipaths $N_{tap}$	9
CP Length $N_{cp}$	8
Relative Speed	360 km/h
Speed of Light $c_{0}$	$3\times 10^{8}$ m/s
Modulation Mapper	QPSK

## Data Availability

Not applicable.

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
