# Peer review of "Transformer-Based Detection for Highly Mobile Coded OFDM Systems"

_entropy, 2023, doi:10.3390/e25060852_

Round 1
Reviewer 1 Report
The article's authors propose an algorithm to decode coded OFDM transmission. Advantages of the article: 1) A novel machine learning-based methodology to decipher OFDM transmissions is given. 2) The authors have implemented the algorithm. 3) The authors tested the performance of the algorithm in simulations. Disadvantages of the article: 1) Figure 5 shows that the algorithm’s performance is worse than the conventional one for uncoded transmission. The hypothesis is that machine learning-based decoding of LDPC codes is more efficient than traditional based on Forney’s graph. But the idea needs an investigation. 2) The authors checked the performance in simulation only. No actual field measurements were made.
OK
Reviewer 2 Report
This paper applies transformer for equalizing an OFDM symbol under doubly selective channel. Using Transformer in wireless communications is a timely topic. And the simulation results demonstrate the effectiveness of the proposed design. There are some technical concerns that need to be addressed before this paper can be considered acceptance.
1. Why only consider replacing equalization part with transformer ? Why not end-to-end design ? Why still need belief propagation for decoding ? Please add comments in the revised version.
2. In the literature review, only one application of transformer in wireless communications is mentioned. In fact, transformer and attention have been applied to other wireless applications, e.g., [R1][R2]. Please provide a more comprehensive review.
3. After (11), how can one know \sigma_i^2 ? If MMSE equalization is used, the noise term in (10) would also contain residual ICI.
4. In Fig. 5, each SNR is trained with a dedicated network, or only one SNR is used during training. If it is the former case, how can one know what SNR the system is operating at in practical scenario ?
[R1] ``Heterogeneous transformer: A scale adaptable neural network architecture for device activity detection," IEEE TWC , May 2023.
[R2] ``Learning to Construct Nested Polar Codes: An Attention-Based Set-to-Element Model," IEEE CL, Dec. 2021.
Other minor comments:
· In (5), should it be f ( x | y,H_f )?
· Before (7), “…. is obtained. The process ….” should be “….. is obtained, the process …”
· In (9), there should be _f for H ?
Nil.
Round 2
Reviewer 1 Report
All my doubts are addressed.
English is OK
Reviewer 2 Report
This version addressed my comments satisfactorily. And it can be accepted in the current form.